

# *Ixodes scapularis* microbiome correlates with life stage, not the presence of human pathogens, in ticks submitted for diagnostic testing

Joshua C. Gil[1], Zeinab H. Helal[2], Guillermo Risatti[2] and Sarah M. Hird[1,3]

[1] Department of Molecular and Cell Biology, University of Connecticut, Storrs, CT, United States of America
[2] Pathobiology and Veterinary Medicine, University of Connecticut, Storrs, CT, United States of America
[3] Institute for Systems Genomics, University of Connecticut, Storrs, CT, United States of America

## ABSTRACT

Ticks are globally distributed arthropods and a public health concern due to the many human pathogens they carry and transmit, including the causative agent of Lyme disease, *Borrelia burgdorferi*. As tick species' ranges increase, so do the number of reported tick related illnesses. The microbiome is a critical part of understanding arthropod biology, and the microbiome of pathogen vectors may provide critical insight into disease transmission and management. Yet we lack a comprehensive understanding of the microbiome of wild ticks, including what effect the presence of multiple tick-borne pathogens (TBPs) has on the microbiome. In this study we chose samples based on life stage (adult or nymph) and which TBPs were present. We used DNA from previously extracted *Ixodes scapularis* ticks that tested positive for zero, one, two or three common TBPs (*B. burgdorferi, B. miyamotoi, Anaplasma phagocytophilum, Babesia microti*). We produced 16S rRNA amplicon data for the whole tick microbiome and compared samples across TBPs status, single vs multiple coinfections, and life stages. Focusing on samples with a single TBP, we found no significant differences in microbiome diversity in ticks that were infected with *B. burgdorferi* and ticks with no TBPs. When comparing multiple TBPs, we found no significant difference in both alpha and beta diversity between ticks with a single TBP and ticks with multiple TBPs. Removal of TBPs from the microbiome did not alter alpha or beta diversity results. Life stage significantly correlated to variation in beta diversity and nymphs had higher alpha diversity than adult ticks. *Rickettsia*, a common tick endosymbiont, was the most abundant genus. This study confirms that the wild tick microbiome is highly influenced by life stage and much less by the presence of human pathogenic bacteria.

Corresponding authors
Joshua C. Gil, joshua.gil@uconn.edu
Sarah M. Hird, sarah.hird@uconn.edu

## INTRODUCTION

Ticks are blood-feeding arachnids. They are globally distributed and are a global public health threat, as they transmit the most diverse set of human pathogens of any known arthropod (*Wikel, 2018*). Common tick-borne pathogens (TBPs) cause: Lyme disease

(also known as Lyme borreliosis, caused by *Borrelia spp.*), anaplasmosis (*Anaplasma phagocytophilum*), Rocky Mountain spotted fever (*Rickettsia rickettsii* and *Rickettsia parkeri*), Texas cattle fever (*Babsia bigemina*), and babesiosis (*B. microti*). The number of tick borne illnesses in the United States continues to increase; with 22,527 cases reported in 2004 and 59,349 cases in 2017 (*Kugeler et al., 2015*). As tick ranges expand north due to climate change (*Dahlgren et al., 2016*; *Wikel, 2018*), the number of tick-associated human illnesses is also expected to increase.

Ticks, like most animals, are host to a variety of microbes that collectively comprise the microbiome. Variation in microbiome composition and structure is linked to properties of the host, such as genetics, physiology and ecology, as well as the environment; however, the relative contributions of these factors varies greatly across hosts. Arthropod microbiomes aid in development, reproduction, immune functioning, digestion of food, and more (*Moran, McCutcheon & Nakabachi, 2008*; *Engelstädter & Hurst, 2009*; *Oliver et al., 2010*; *Ferrari & Vavre, 2011*; *Cordaux, Bouchon & Grève, 2011*; *Wernegreen, 2012*). The gut microbiome can also act as a defense for the host, in both vertebrates and invertebrates, by competitive exclusion of pathogens, activation of host immune responses and secretion of inhibitory secondary metabolites (*Bonnet et al., 2017*; *Saldaña, Hegde & Hughes, 2017*; *Pickard et al., 2017*).

Arthropod microbiome transmission occurs both vertically, whereby bacteria are passed from mother to eggs, as well as horizontally, from the environment (*Bonnet et al., 2017*). Bacteria that are transmitted vertically are often endosymbionts (intracellular bacteria that cannot survive outside of the host's cells) and aid in the host's fitness and health (*Bonnet et al., 2017*). Like many insects with nutrient poor diets, ticks require endosymbionts for micronutrient synthesis because they are unable to fully digest blood and synthesize all the essential nutrients independently (*Bonnet et al., 2017*). In *I. scapularis*, a hard bodied tick commonly referred to as deer tick, the most frequently observed endosymbiont is *Rickettsia*, which can range in relative abundance from 40% to 90%, depending on geographic location and life stage of the tick (*Benson et al., 2004*; *Van Treuren et al., 2015*; *Thapa, Zhang & Allen, 2019*). Ticks develop through three life stages, first as larva, then nymph, and finally adult. In each life stage, *I. scapularis* will only feed once limiting the amount of bacteria ingested via diet; however, ingesting a meal once at each life stage does not preclude ticks from having diverse microbiomes. Ticks do not have a stable microbiome and can vary significantly depending on what animal they get there blood meal (*Ross et al., 2018*; *Landesman et al., 2019a*). The microbiome of arthropods can likewise be altered by which endosymbionts are present, as well as provide protection against invasive pathogens (*Łukasik et al., 2013*; *Hendry, Hunter & Baltrus, 2014*; *Abraham et al., 2017*).

The role of TBPs within the tick microbiome is poorly understood. *Ixodes scapularis* is the primary vector for transmitting *B. burgdorferi* but can also transmit *A. phagocytophilum*, *B. miyamotoi,* and *B. microti*. Variation in *Ixodes* microbiome has been correlated to the tick's life stage, local environment, and the specific members of the host microbiome (*Hawlena et al., 2013*; *Narasimhan & Fikrig, 2015*; *Van Treuren et al., 2015*; *Zolnik et al., 2016*; *Abraham et al., 2017*; *Kwan et al., 2017*; *Thapa, Zhang & Allen, 2019*). In *I. pacificus*, as ticks develop from larva to adult, microbiome richness and evenness decrease (*Kwan*
*et al., 2017*). In *I. scapularis,* increased microbiome diversity is associated with increased colonization of both *B. burgdorferi* and *B. microti.* In addition, nymph beta diversity is correlated with the presence/absence of *B. burgdorferi* (*Landesman et al., 2019b*). Another TBP, *A. phagocytophilum*, alters the microbiome in *I. scapularis* by promoting its own growth at the expense of other microbes (*Abraham et al., 2017*). *A. phagocytophilum* decreases the peritrophic matrix found between the tick's lumen and epithelium which promotes its establishment while impeding colonization by other bacteria, including *B. burgdorferi* (*Abraham et al., 2017*). In these studies, ticks were captive and infected with a single TBP; in nature, it is not uncommon to find multiple pathogens in the same tick (*Steiner et al., 2008*).

To elucidate the natural biodiversity of the *I. scapularis* microbiome, we used ticks that were voluntarily sent to the Connecticut Veterinary Medical Diagnostic Laboratory at the University of Connecticut for pathogen detection. We analyzed total extracted DNA from ticks from two life stages (nymph and adult) and with multiple combinations of known human pathogens. Our goals were to see if we can attain biologically relevant data with a collaboration in the Department of Pathobiology and Veterinary Science. Using ticks sent in from people for pathogen detection across Connecticut, we aimed to determine if microbiome composition or diversity changes are correlated to (1) pathogen presence, including in ticks with multiple TBPs and (2) tick life stage.

## MATERIALS & METHODS

### Morphological identification of ticks

Ticks used in this study were sent to Connecticut Veterinary Medical Diagnostic Laboratories (CVMDL) by Connecticut residents from across the state and the precise locations where the ticks were acquired by the humans are not available. The ticks used in this study were submitted to CVMDL in spring and summer of 2017 and 2018 ($n = 31$ and $n = 14$ respectively). They are removed prior to submission and are either dropped off directly at the laboratory or sent by mail. All ticks in this study were processed by the CVMDL $\sim$3–4 days post collection. Ticks submitted to CVMDL for diagnostic testing purposes were morphologically identified according to identification keys by *Keirans & Litwak (1989)* and US National Tick Collection (https://cosm.georgiasouthern.edu/usntc/) using a stereo microscope LAXCO LMSP-Z 230P (Mill Creek, WA) under power of magnification ranging from 6.7 to 45X.

Ticks were then classified based on engorgement (non-engorged, slightly, partial, fully); to control for variation from engorgement status we preferentially chose ticks that were either slightly or partially engorged (*Ross et al., 2018*). Adult ticks were sexed visually, but the nymphs were not, as nymphs cannot be sexed visually. All samples were extracted on the day they arrived at CVMDL.

### Extraction of total DNA from tick specimens

DNA was extracted from individual ticks using a Nucleospin Tissue kit (cat# 740952-250, Macherey-Nagel GmbH & Co. KG, PA). Briefly, ticks were placed in 180 μL of buffer T1or in 360 μL of the same buffer for engorged specimens. Sterile sand was then added

**Table 1  Primers used for amplification of specific pathogens in tick specimens.**

| Pathogen-References | Sense | Primer sequence (5′–3′) | Gene targeted |
|---|---|---|---|
| *Borrelia burgdorferi sensu lato* | F | GTGGATCTATTGTATTAGATGAGGCTCTCG | *rec A* |
| *Pietilä et al. (2000)* | R | GCCAAAGTTCTGCAACATTAACACCTAAAG | |
| *Anaplasma phagocytophilum* | F | CCAGCGTTTAGCAAGATAAGAG | *msp2* |
| *Pesquera et al. (2015)* | R | GCCCAGTAACAACATCATAAGC | |
| *Babesia microti* | F | CTTAGTATAAGCTTTTATACAGC | *ssrRNA* |
| *Adelson et al. (2004)* | R | TAGGTCAGAAACTTGAATGATACA | |
| *Borrelia miyamotoi* | F | ATAGCTCACAGGGGTGC | *qlpQ* |
| (CVMDL in house) | R | CTCGATTGGGAAATAATTGTGC | |

to the tube and tick bodies were manually homogenized using sterile disposable wood applicators. After homogenization, 25 µl of proteinase K were added to the mix followed by an incubation step at 56 °C for 4 h. The DNA extraction procedure proceeded then as recommended by the manufacturer. Positive extraction control (PEC) tick was included with each tick DNA extraction, the PEC tick was selected to be free from all tested pathogen. Ticks are processed and tested as they are received; including negative extraction controls on diagnostic samples is not in the standard protocol, as it is not a significant source of error for pathogen detection. To detect potential contamination from extraction, we followed *De Goffau et al. (2018)* and tested for batch variation based on dates of extraction. We also confirmed via a thorough literature search that the taxa detected in our samples could reasonably have come from ticks (i.e., are ecologically plausible).

## qPCR for detection of pathogens

SYBR green-based qPCRs were used to detect genomic DNA of pathogens (see Table 1 for primers used and references). Fast SYBR Green Master Mix (cat# 4444432, Applied Biosystems, Foster City, CA) was used in 25 µl reaction following manufacturer's recommendations. DNA was amplified using an ABI 7500 Fast Real-Time PCR thermal cycler (Applied Biosystems, Foster City, CA) by an initial denaturation step hold at 95 °C for 3 min followed by 45 cycles at 95 °C for 30 s, 60 °C for 30 s, and 72 °C for 30s. Signal was captured at each cycle at the end of elongation steps. Melting curve analysis was performed at the end of each run for 95 °C for 15s, 60 °C for 1 minute, 95 °C for 15s and 60 °C for 15s. For each real time PCR run, a Non-Template Control (NTC) was used. The NTC produced negative result for each run showing there was no external contamination. A Positive Amplification Control was used in each run as well. A sample is considered positive for a particular pathogen when an amplification curve is observed (Ct value obtained). Each sample was tested for *B. burgdorferi, A. phagocytophilum, B. microti,* and *B. miyamoti*, see Table S1 for each sample's pathogen Ct scores.

## Sample selection and 16S rRNA gene sequencing

We selected samples with between zero and three common TBPs (Table 2). One adult tick had three TBPs (*A. phagocytophilum, B. burgdorferi,* and *B. microti,* referred to as ''[Bb, Ap, Bab]''), seven had two TBPs (three adults: *B. burgdorferi, A. phagocytophilum,* ''[Bb,

**Table 2  Tick-borne pathogen status, notation used throughout paper and sample sizes.**

| TBP(s) conditions tested | Notation | Sample Size | Nymphs | Adults |
|---|---|---|---|---|
| *B. burgdorferi* | [Bb] | 13 | 6 | 7 |
| *A. phagocytophilum* | [Ap] | 7 | 3 | 4 |
| *B. miyamotoi* | [Bm] | 4 | 2 | 2 |
| *B. microti* | [Bab] | 1 | 0 | 1 |
| *B. burgdorferi, A. phagocytophilum* | [Bb, Ap] | 5 | 2 | 3 |
| *B. burgdorferi, B. microti* | [Bb, Bab] | 2 | 0 | 2 |
| *B. burgdorferi, A. phagocytophilum, B. microti* | [Bb, Ap, Bab] | 1 | 0 | 1 |
| No TBP detected | [TBP-] | 10 | 5 | 5 |

Ap]", two adults: *B. burgdorferi, B. microti* "[Bb, Bab]", two nymphs: *A. phagocytophilum, B. burgdorferi,* "[Bb, Ap]"), 25 had one TBP (Table 2). Negative sequencing controls were used to verify the lack of contaminants in reagents used during amplification and sequencing.

   All of the adults used for this study were females and the sexes of the nymphs are unknown. We were unable to precisely control the engorgement levels; however, most ticks selected were either partially or slightly engorged. One was engorged (adult) and two were not engorged at all (one nymph and one adult).

   DNA extracts were amplified and sequenced at the University of Connecticut Microbial Analysis, Resources and Services center using the standard protocol for amplification and sequencing. Extracts were quantified using the Quant-iT PicoGreen kit (Invitrogen, ThermoFisher Scientific). Partial bacterial 16S rRNA genes (V4, 0.8 picomole each 515F and 806R with Illumina adapters and 8 basepair dual indices (*Kozich et al., 2013*) were, in triplicate, amplified in 15 ul reactions using GoTaq (Promega) with the addition of 10 µg BSA (New England BioLabs). We added 0.1 femtomole 515F and 806R which does not have the barcodes and adapters to overcome initial primer binding inhibition, because the majority of the primers do not match the template priming site. The PCR reaction was incubated at 95 ° C for 2 minutes, the 30 cycles of 30 s at 95.0 °C, 60 s at 50.0 °C and 60 s at 72.0 °C, followed by final extension as 72.0 °C for 10 minutes. PCR products were pooled, quantified and visualized using the QIAxcel DNA Fast Analysis (Qiagen). PCR products were pooled using QIAgility liquid handling robot after the products were normalized based on the concentration of DNA from 350-420 bp. The pooled PCR products were cleaned using the Mag-Bind RxnPure Plus (Omega Bio-tek) according to the manufacturer's protocol. The cleaned pool was sequenced on the Illumina MiSeq using v2 $2 \times 250$ base pair kit (Illumina, Inc). Three PCR controls were also sequenced to test for PCR reagent contamination.

## Sequence processing

All data analyses and sequence processing were done using R v.3.5.0 (*R Core Team, 2019*). Sequences were quality controlled, denoised and merged using DADA2 (*Callahan et al., 2016*) to create a sample by ASV (amplicon sequence variant) matrix. An ASV is an

operational taxonomic unit, defined as any unique sequence that passes stringent quality control. Taxonomy of ASVs was assigned using RDP's Naïve Bayesian Classifier with the Silva reference database v128 (*Wang et al., 2007*; *Quast et al., 2013*). Sequences that were identified as mitochondria, chloroplasts, or that were unable to be confidently assigned to any bacterial phylum were removed. We also removed 22 ASVs that were identified as *Candidatus* Carsonella, an endosymbiont found only in psyllids (*Sloan & Moran, 2012*). We removed these ASVs because they were nested within the mitochondria clade in our original analysis. We further verified that these 22 ASVs were likely mitochondrial in origin using blastn. We conducted a filtered search, only keeping matches with >90% sequence identity and e-value < 1e-40. We found 15 ASVs were not assigned to *Candidatus Carsonella* but instead, matched best with *I. scapularis* and *I. pavlovskyi* mitochondrial sequences see supplemental Table S2 (*Zhang et al., 2000*; *Morgulis et al., 2008*). The remaining 7 ASV sequences could not be assigned to any organism using the previous search parameters.

To remove likely contaminants, we processed the sequences using the Decontam package (*Davis et al., 2018*), which uses the negative controls to identify likely contaminants.

To calculate phylogenetic diversity metrics, we performed a multiple-alignment of all ASVs using the *DECIPHER* package in R (*Wright, 2015*), and constructed a phylogenetic tree with the *phangorn* package version 2.4.0 (*Schliep, 2011*).

## Data analyses

Data analysis was done using the R packages phyloseq (*McMurdie & Holmes, 2013*), vegan (*Oksanen et al., 2007*), and DESeq2 (*Love, Huber & Anders, 2014*). Code was generated from vegan and phyloseq tutorials available online. We rarefied samples to an even depth of 8,252 reads for alpha and beta diversity analyses, which resulted in the loss of two samples from the dataset for being below 8,252 reads. See Fig. S1 for rarefaction curves.

Shannon diversity index was used to quantify the alpha diversity of the samples. Using a Shapiro test, we determined the data were not normally distributed; therefore, we tested the significance of life stage of ticks using a Kruskal–Wallis test. The alpha diversity was first tested on the rarefied data; however, low biomass samples have the potential to artificially inflate diversity (*Salter et al., 2014*; *Erb-Downward et al., 2020*). To address this, we removed ASVs that comprised less than 0.1%, 1%, 5%, and 10% of the overall sequence count in each sample, rarefied again to 8200 reads, and tested alpha diversity using the Kruskal–Wallis test between adults and nymphs (*Couper et al., 2019*). We conducted multiple pairwise comparisons of alpha diversity looking at different TBP conditions using Wilcoxon rank-sum test.

Beta diversity was assessed by generating pairwise distance matrices of Bray–Curtis, weighted Unifrac, unweighted Unifrac, and Jaccard distances (*Lozupone et al., 2007*; *Xia & Sun, 2017*). We plotted NMDS ordinations using phyloseq and tested for significance of the metadata using a Permutational multivariate analysis of variance (PERMANOVA). We tested how the different TBPs correlate to the microbiome. The data were tested by comparing the samples when grouped by their combination of TBPs, refer to Table 2 for TBP combinations. The samples were also categorized by whether they had one of the four pathogens regardless of whether that sample had additional TBPs. We did this

in order to see if one TBP correlated to significant variation in the microbiome. Beta diversity was tested using PERMANOVA. We tested life stage by merging the [TBP-] and [Bb] samples when we found no significant difference in those samples' microbiome, we did this to increase the sample size of both nymphs and adults. Furthermore, we tested for temporal effects on the microbiome and tested changes in beta diversity using the previously mentioned distance metrics. We tested adults and nymphs separately using the whole data set with sequences <1% removed and rarefied to 8200 reads.

To test if TBPs could influence diversity results, we tested using the same methods as before, but we removed all TBP reads and rarefied to a new sequencing depth (3,204 reads). We then tested these results as described previously. To confirm there was no difference, we rarefied the original data set to 3,204 reads with the TBP still present and ran a PERMANOVA on those data again to ensure results were consistent across rarefication depths.

We identified differential abundance of genera from different life stages and different TBPs using DESeq2 (v. 1.24.0) in R. Genera were identified as differentially abundant if the corrected $p < 0.05$. $p$-values were corrected with the Benjamini and Hochberg false discovery rate for multiple testing (*Benjamini & Hochberg, 1995*) .

Relative abundance of taxa in our samples were calculated after the samples had been rarefied (8,252 reads). Low abundance taxa in each sample were grouped if they totaled <1% of the reads at the phylum level and <15% of the genera. The relative abundances for each taxon in each sample were averaged to get the relative abundance for each TBP condition and life stage.

## RESULTS

### Sequence data and homogeneity check

The initial dataset had 2,449,198 sequences and the final dataset had 2,114,742 high-quality sequences. Sequences per sample ranged from 5,018–126,549 (mean = 46,994). No sequences were identified as likely contaminants.

Homogeneity of variance of the rarefied data was tested using ANOVA and Bray–Curtis distances. We saw no significant difference in life stage or pathogen status (life stage ANOVA: F 1, 4 1 = 0.4685, $p = 0.4975$; pathogen status ANOVA: F 7, 3 5 = 2.1431, $p = 0.06442$).

### *Ixodes scapularis* microbiome: bacterial composition

The mean relative abundances of the phyla dominating the [TBP-] samples were Proteobacteria (78.9%), Firmicutes (10.89%), and Spirochaetes (7.4%). [Bb] ticks were similar to [TBP-]: Proteobacteria comprised the majority (77.65%), Firmicutes (14.26%), and Spirochaetes (6.5%). [TBP-] samples had 8 phyla that were >1% abundances in at least one sample. [Bb] had five phyla that were >1% of the relative abundance (Table S3). The genera in both [TBP-] and [Bb] were largely similar and dominated by *Rickettsia* (43.3% and 43.1% respectively), Fig. 1. [TBP-] and [Bb] had almost equal amounts of rare taxa (relative abundance < 15%): 19.4% and 18.7%, respectively. Complete table relative abundance of genera in [TBP-] and [Bb] samples available, see supplemental Table S4.

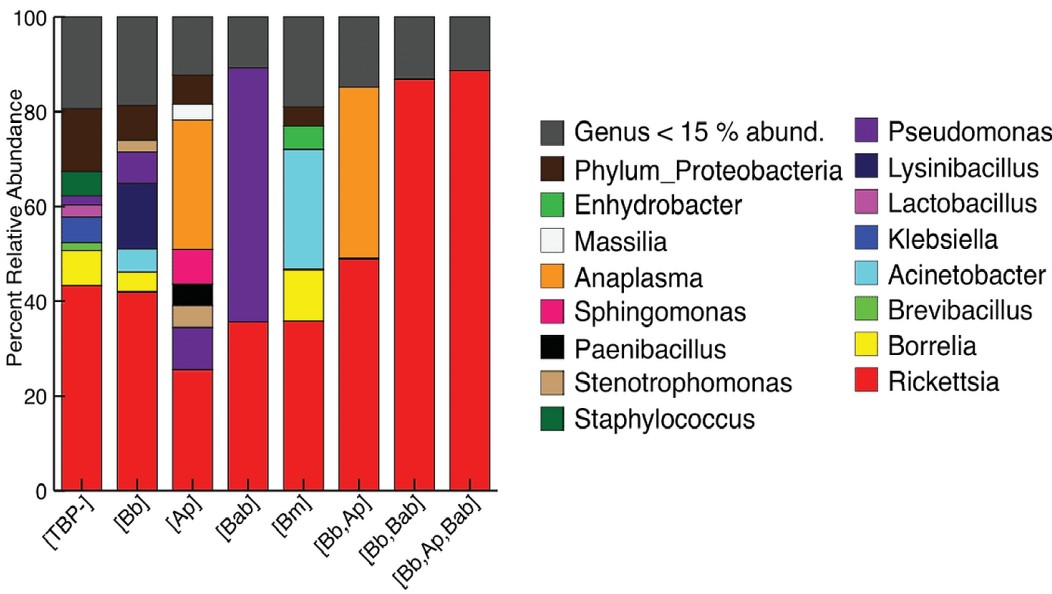

**Figure 1  Taxonomy graph showing average relative abundance of genera in each TBP condition.** Genera less than 15% of the relative abundance in each sample were grouped into a new category. ([TBP-]; $n = 10$, [Bb]; $n = 13$, [Ap]; $n = 7$, [Bab]; $n = 1$, [Bm]; $n = 4$, [Bb,Ap]; $n = 5$, [Bb,Bab]; $n = 2$, [Bb,Ap,Bab]; $n = 1$).

## Effect of *B. burgdorferi* infection in *I. scapularis* microbiome composition

For clarity, the first analyses were between just the [TBP-] and [Bb] ticks. Alpha diversity was not significantly different between [Bb] ($n = 13$) and [TBP-] ($n = 10$), Shannon: $X^2 = 0.0038462$, $df = 1$, $p = 0.9505$, (Fig. 2A). NMDS ordinations showed minimal clustering of samples by the presence of *B. burgdorferi* (Fig. 2B). Infection status was not significant ($p > 0.05$) using three of the four distance metrics, although infection status explained 8% of the variation in the unweighted Unifrac NMDS ($df = 1$, $R^=0.08$, $p = 0.024$, Table 3).

## TBPs, single and co-infections effect on the microbiome

We next included all samples, including [TBP-], single infection and coinfected ticks. The most abundant phylum, regardless of TBP status, was Proteobacteria (77.7–98.2%). Firmicutes ranged from 1.5–14.3%, except for in [Bb, Ap, Bab] and [Bb, Bab], where no Firmicutes were present above 1% of the relative abundance. Spirochaetes were in all samples except TBP statuses [Ap] and [Bab]; in the remaining TBP statuses, Spirochaetes ranged from 1.1–13.79% (Table S3). *Rickettsia* was the most abundant genera in all TBP statuses except for [Ap] and [Bab]. [Ap] had similar levels of *Rickettsia* and *Anaplasma* (25.4% and 27.2% respectively). In [Bab], *Pseudomonas* comprised 53.7% of the microbiome. *Anaplasma* was also in high abundance in [Bb, Ap] with *Rickettsia* (48.9%) and *Anaplasma* (36.2%). Only [Ap] and [Bb, Ap] had *Anaplasma* with relative
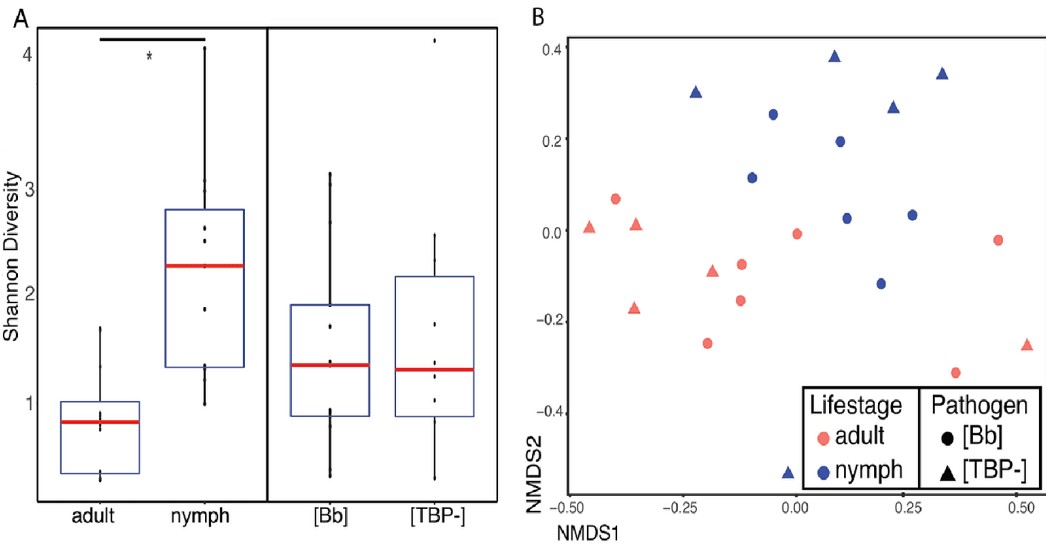

**Figure 2** **Alpha and beta diveristy of adults vs nymphs and [Bb] vs [TBP-] ticks.** (A) Shannon diversity boxplot. (*) denotes significant difference in Shannon diversity between adults ($n = 12$) and nymphs ($n = 11$). Significance was determined by the Kruskal–Wallis test ($p = 0.005$). (B) Bray–Curtis NMDS ordination showing clustering of adults (red) and nymphs (blue). No significant clustering in [Bb] (circles) or [TBP-] (triangles).

abundance greater than 15%. *Borrelia* average abundance in [Bb] was 4.2%, [Bm] 10.9%, and [TBP-] 7.4% (Fig. 1).

The number of TBPs in a tick did not correlate with beta diversity in *I. scapularis*. We compared samples that had one, two or three TBPs present and regardless of distance metric, there was no significant correlation between infection status (the number of TBPs) and the microbiome (Bray–Curtis: $df = 7$, $R^2 = 0.15833$, $p = 0.586$; weighted Unifrac: $df = 7$, $R^2 = 0.1545$, $p = 0.598$; unweighted Unifrac: $df = 7$, $R^2 = 0.19906$, $p = 0.127$; Jaccard: $df = 7$, $R^2 = 0.15858$, $p = 0.604$). To see if the presence of the pathogens themselves were altering the diversity of the microbiome, we removed the TBPs ASVs and beta diversity was still not significantly correlated to TBPs (Table 4). Bray-Curtis NMDS ordinations with and without TBPs ASVs showed minimal clustering (Figs. 3A–3B). No single TBP had a significant effect on the beta diversity in *I. scapularis* (Table 4), except *B. burgdorferi,* which was significant when using unweighted Unifrac ($R^2 = 0.06849$, $p = 0.002$) (Table 4).

Shannon diversity indices indicated no significant difference in alpha diversity of any of the TBP statuses ($p > .05$) with the exception of [Bb, Ana] and [Bm] (Wilcoxon rank-sum: $p = 0.032$) (Fig. 4). We then separated samples into either [Bb], [Ap], [Bm], or [Bab] regardless of multiple TBPs and we found there were no differentially abundant genera. The only genera that were differentially abundant in the samples were the TBPs in question. Ticks classified as [Bb] or [Bm] had *Borrelia* significantly more abundant. Similarly, [Ap] ticks had significantly more *Anaplasma*. [Bab] is a eukaryote and hence there were no *Babesia* sequences to test.

**Table 3  Effect size ($R^2$) and significance ($p$) of *B. burgdorferi* infection and life stage using PERMANOVA.** Comparisons with $p < 0.05$ are bolded.

| Distance metric | [Bb] and [TBP-] | | Life stage | |
| --- | --- | --- | --- | --- |
| | $R^2$ | $p$ | $R^2$ | $p$ |
| Bray–Curtis | 0.04541 | 0.395 | **0.15164** | **0.004** |
| Weighted Unifrac | 0.01953 | 0.728 | **0.15781** | **0.006** |
| Unweighted Unifrac | **0.08098** | **0.024** | **0.11624** | **0.002** |
| Jaccard | 0.04278 | 0.465 | **0.14973** | **0.005** |

**Table 4  Different TBP ASVs effect on beta diversity of *I. scapularis* microbiome using PERMANOVA.** Different distance metrics with statistically significant $p$-values ($p < 0.05$) are in bold. Affect size ($R^2$) of significant $p$-value are in bold.

| | Bray–Curtis | | Weighted Unifrac | | Unweighted Unifrac | | Jaccard | |
| --- | --- | --- | --- | --- | --- | --- | --- | --- |
| | $R^2$ | $p$ | $R^2$ | $p$ | $R^2$ | $p$ | $R^2$ | $p$ |
| TBPs present | 0.15833 | 0.586 | 0.1545 | 0.598 | 0.19906 | 0.127 | 0.15858 | 0.604 |
| TBPs absent | 0.13587 | 0.796 | 0.09385 | 0.862 | 0.15629 | 0.626 | 0.14397 | 0.771 |
| *B. burgdorferi* | 0.03455 | 0.120 | 0.0134 | 0.631 | **0.06849** | **0.002** | 0.03179 | 0.138 |
| *A. phagocytophilum* | 0.03312 | 0.157 | 0.02246 | 0.443 | 0.03895 | 0.053 | 0.02947 | 0.175 |
| *B. miyamotoi* | 0.02539 | 0.305 | 0.01408 | 0.530 | 0.02203 | 0.484 | 0.02353 | 0.385 |
| *B. microti* | 0.02473 | 0.277 | 0.01126 | 0.681 | 0.01281 | 0.964 | 0.02404 | 0.331 |

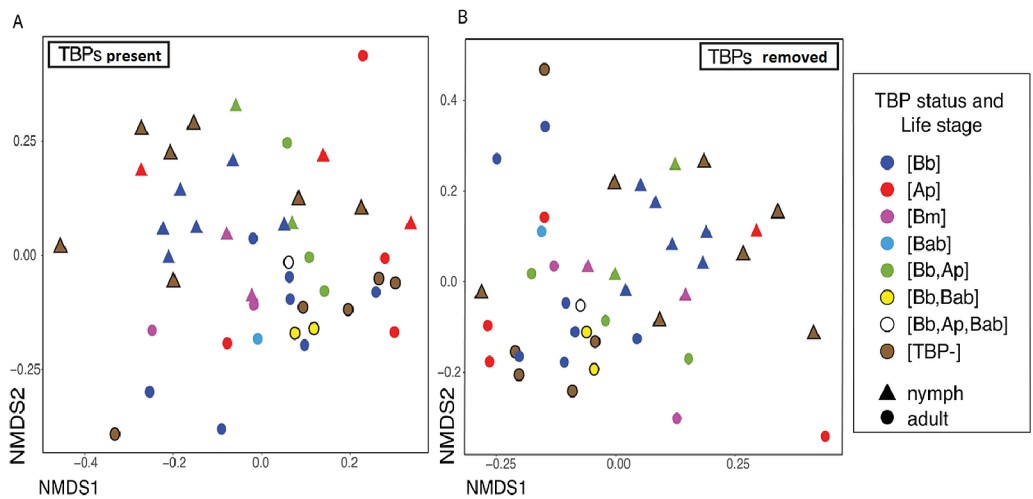

**Figure 3  Different TBPs effect on microbiome beta diveristy.** Ordinations of whole data set rarefied to 3,200 reads to maintain all 43 samples when the TBPs ASVs are removed (A) Bray–Curtis distance NMDS ordination with TBP reads present in the samples. (B) Bray–Curtis Distance NMDS ordination with TBP reads removed from each sample.

There was no significant correlation ($p > 0.05$) between the month and year that the tick was collected and beta diversity except when using Jaccard distance with the nymphs ($df = 8$, $p = 0.027$, $R^2 = 0.56164$).

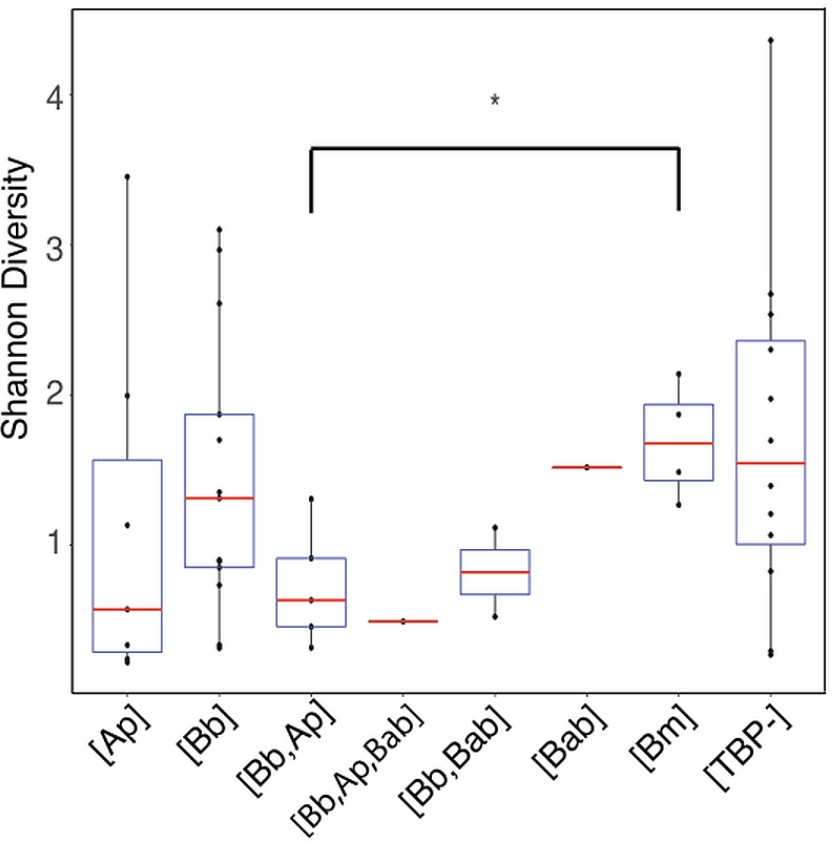

**Figure 4   Shannon diversity boxplot comparing TBPs.** Significant difference in alpha diversity were determine by the Wilcoxon rank-sum test. Significant differences are denoted by (*) and was only observed in [Bb,Ap] and [Bm] ($p = 0.032$). [TBP-]; $n = 10$, [Bb]; $n = 13$, [Ap]; $n = 7$, [Bab]; $n = 1$, [Bm]; $n = 4$, [Bb,Ap]; $n = 5$, [Bb,Bab]; $n = 2$, [Bb,Ap,Bab]; $n = 1$.

## Microbiome diversity and *I. scapularis* life stage

*Ixodes scapularis* life stage correlates with microbiome composition. Alpha diversity differed significantly between adults and nymphs (Kruskal–Wallis, Shannon index: $X = 11.978$, $df = 1$, $p = 0.0005384$, Fig. 2A). Shannon diversity index in adults was $0.843 \pm 0.148$ and was $2.241 \pm 0.305$ in nymphs. Accounting for artificial inflation of alpha diversity in low biomass samples; adults and nymphs were still significantly different when low abundance ASVs (less than 0.1% or 1%) were removed. When ASVs that comprised less than 5%, and 10% of the total ASVs were removed, we no longer saw significant differences between life stages (Table S5). Life stage was significantly correlated with Bray–Curtis, weighted Unifrac, unweighted Unifrac, and Jaccard distance metrics (Table 3). Differences in life stage can be visualized in Bray–Curtis NMDS ordination (Fig. 2B). Life stage explained between 11–15% of the variation observed (Table 3).

Nymphs had eight phyla >1% relative abundance: Proteobacteria (68.0%), Firmicutes (14.8%), and Spirochaetes (13.5%) accounted for most of the communities. Adults had four phyla with >1% relative abundance, with the largest being Proteobacteria (87.5%) and

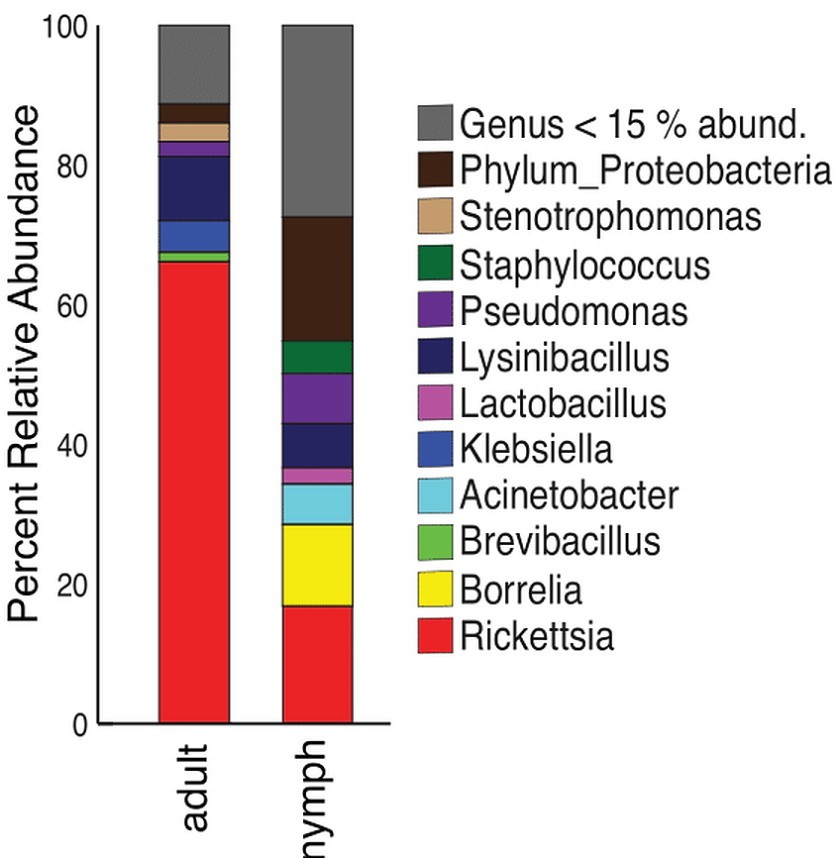

**Figure 5  Taxonomy graph showing the average relative abundance of genera in adults and nymphs.** Genera less than 15% of the relative abundance in each sample were grouped into a new category. Adults ($n = 12$) and nymphs ($n = 11$).

Firmicutes (10.9%) (Table S6). When averaged, adult ticks had seven genera with relative abundance >15%; whereas, nymphs had eight. *Rickettsia* was more abundant in adults than in nymphs (66.1% vs 16.8%). The averaged relative abundance of genera in nymphs was not dominated by a singular taxa. The most abundant genera identified in nymphs belonged to an unknown genus in Proteobacteria (17.7%), *Rickettsia* (16.8%), and *Borrelia* (11.7%) (Fig. 5). *Rickettsia* was absent in three adult samples; the rest had *Rickettsia* present at higher than 74.2%. The three adult samples that did not have *Rickettsia* had a similarly large amount of *Lysinibacillus* (two samples 46.0% and 64.3%) or *Klebsiella* (54.3%). Half of the nymphs had *Rickettsia*; one sample was similar to adults (82.6% relative abundance) the other five ranged from 14.7–28.7%. Five nymphs did not have *Rickettsia* and have different genera predominating (Table S7).

## DISCUSSION

The microbiome of pathogen vectors may provide critical insight into disease transmission and management (*Bonnet et al., 2017*). *Ixodes* ticks can spread numerous human pathogens that are responsible for thousands of cases of illness each year. Lyme disease in particular

is increasingly prevalent, potentially due to human environmental disturbances. A better understanding of the microbiome and its relationship to human pathogens is important for understanding, and perhaps even identifying, possible candidates for microbial controls for tick-borne pathogen (TBP) (*Saldaña, Hegde & Hughes, 2017*). Wild ticks are needed to characterize the natural relationship between the host, the microbiome and TBPs. In this study, we tried to derive ecologically relevant data in collaboration with a diagnostic laboratory, using DNA extracts derived from pathogen detection services to investigate the relationship between microbiome, life stage and TBPs found in wild *I. scapularis*. In particular, we wanted to assess how different TBPs (*B. burgdorferi*, *A. phagocytophilum*, *B. miyamotoi*, and *B. microti*) and how multiple TBPs in a single tick may influence the microbiome.

The most common TBP, *B. burgdorferi*, had minimal effect on the overall diversity and composition of the microbiome of *I. scapularis*. [Bb] and [TBP-] microbiomes were similar in alpha and beta diversity with one exception: [Bb] and [TBP-] ticks were significantly different ($p = 0.024$) using unweighted Unifrac distance. Unweighted Unifrac distance is a metric based on the presence of phylogenetic lineages within a microbiome and may be more sensitive to rare taxa than other distance metrics; in this case [TBP-] happens to have less abundant but more unique ASVs compared [Bb] (191 ASVs in [TBP-] vs 96 ASVs [Bb]). Our results indicate that communities with *B. burgdorferi* are not significantly different from those with no TBPs. These findings are congruent with results in *I. pacificus*, where the microbiome was not affected by *B. burgdorferi* (*Kwan et al., 2017*).

Similar to *B. burgdorferi*, mono-infection with other TBPs (*A. phagocytophilum, B. miyamotoi,* and *B. microti*) did not result in significant differences in alpha diversity compared to [TBP-] ticks. Regardless of which TBP was present, no taxa were significantly more or less abundant in the microbiome except for the TBP in question. This suggests that the microbiomes with these TBPs are not significantly different in composition or diversity from [TBP-] ticks. These data suggest there is no difference at the time of extraction; however, it is still possible that the full effects of the TBPs present will not be actualized until the tick has fed completely and been given time for the microbiome to react to the new blood meal. This finding is contrary to results found in lab reared *I. scapularis* infected with *A. phagocytophilum*, where the presence of *A. phagocytophilum* reduced the relative abundance of difference Gram-positive bacteria (*Abraham et al., 2017*) and underscores the importance of including wild animals in microbiome studies (*Hird, 2017*).

Co-infection with multiple TBPs did not have an apparent effect on alpha or beta microbiome diversity; ticks that had one, two, or three TBPs present had similar alpha and beta diversity. To ensure that the TBPs were not obscuring changes in the underlying non-TBP community, we re-analyzed the samples after removing the TBP sequences; the removal of TBP sequences still did not affect beta diversity. The only significant difference among the groups was between [Bb, Ap] and [Bm]. We hypothesize this difference is due to the high abundance of *Anaplasma* in [Bb, Ap] and large abundances of *Acinetobacter* and *Borrelia* in [Bm]. Notably, these results could be partially driven by our low sample sizes for ticks that carried two or three pathogens; further tests on these and additional combinations of TBPs will elucidate the role of TBPs on the tick microbiome. Taken

together, our data suggest that the presence of TBPs does not significantly change the microbiome in wild tick.

Current data on the tick microbiome show two prominent conflicting trends in microbiome diversity regarding the life stage in *Ixodes* spp. In *I. scapularis* microbiome, alpha diversity increased as the tick ages, increasing from larvae to nymph and nymph to adult (*Clay et al., 2008*; *Zolnik et al., 2016*). In contrast, *I. pacificus*, showed decreasing diversity as the tick ages (*Carpi et al., 2011*; *Kwan et al., 2017*). In our study, ticks showed lower alpha diversity in adults compared to nymphs but this result comes with a caveat. The adult ticks in our study were all females. All of our samples were collected after they had started to feed on a human. The adult female ticks in this study may contain *Rickettsia* at high abundances because we used the whole organism and *Rickettsia* is frequently found in the ovaries (*Zolnik et al., 2016*). Another possibility that might explain increased alpha diversity in nymphs is the low initial biomass of the samples, which are physically much smaller than adults leading to decreased bacterial load. Smaller size can inflate alpha diversity (*Salter et al., 2014*; *Erb-Downward et al., 2020*). We used DNA concentration post extraction as a proxy for biomass. Upon visual inspection, nymphs with high input biomass show similar levels of variation as the nymphs with lower concentrations of DNA post extraction (Fig. S2). To further test this, we removed low abundance ASVs from the samples and tested for significant differences in alpha diversity based on life stage. We defined "low abundance" as ASVs totaling less than 0.1%, 1%, 5%, and 10% to determine if the alpha diversity measures in the smaller bodied nymphs was conflated by an increase of rare and low abundant taxa from the cuticle. When we removed ASVs whose sum was less than 0.1% and 1% in each sample there was still a significantly greater alpha diversity in nymphs than in adults. This suggests that higher alpha diversity in nymphs is likely not the result of differences in input bacterial biomass, although this deserves further experimental verification.

*Ixodes scapularis*' microbiome can vary significantly depending on where the tick was sampled (*Van Treuren et al., 2015*). *Rickettsia* is frequently the dominant member of the microbiome as it is the most common endosymbiont found in *Ixodes*; however, this is not always the case (*Varela-Stokes et al., 2017*). *I. scapularis* from the mid-Atlantic have genera from the family Enterobacteriaceae comprising the majority of the taxa and *Rickettsia* is present at significantly lower abundances. Ticks in the northeast United States have more variable microbiomes than those in the mid-Atlantic and most have *Rickettsia* as the dominant member; *Sphingomonas* and *Borrelia* are additional genera at high abundance (*Van Treuren et al., 2015*). We observed *Rickettsia* as the most abundant taxon in 27 of the 44 samples; however, there was significant variability. *A. phagocytophilum* was the most dominant species in four samples (>80%), and in the remaining 13 samples, no single taxon dominated. Instead, they contained a diverse set of genera (*Sphingomonas*, *Stenotrophomonas*, *Pseudomonas*, *Borrelia*, *Lysinibacillus*, and others; see Table S4 for specific relative abundances).

Our data reaffirms that *I. scapularis* from the northeastern United States are highly variable, unlike *I. scapularis* found in other regions of the United States. The variability in abundance and absence of *Rickettsia* in female ticks in the northeastern United States

further suggests that there might be another endosymbiont filling the biological role of *Rickettsia*. For example, *Fransicella*-like endosymbiont, a common endosymbiont in dog ticks, *Dermacentor variabilis*, was found in seven of our samples (*Varela-Stokes et al., 2017*). In the samples with *Fransicella*-like endosymbiont, five had *Rickettsia* and the other two had large quantities of *Anaplasma*. Could one of these taxa be supplying *I. scapularis* with required micronutrients? We found trace amounts (i.e., <1%) of *Wolbachia* in one sample. *Wolbachia* is an endosymbiont in cicadas, but it is a parasite in other arthropods (*Correa & Ballard, 2016*). *Wolbachia* has been found in *I. scapularis* at varying abundances (*Duron et al., 2017*; *Benson et al., 2004*; *Thapa, Zhang & Allen, 2019*). Because we only found one tick in our study to have *Wolbachia* and it was at low abundance, further surveying should be done to determine how common *Wolbachia* is in *I. scapularis* across the northeastern United States populations. These data would resolve the number and diversity of endosymbionts and intracellular parasites present in *I. scapularis*.

All of our samples were collected when found feeding on a person and all but three were either slightly or partially engorged. We were unable to verify if the engorgement was a significant correlate to the microbiome data due to the low sample sizes. However, we visually compared relative abundance data and saw the engorged and non-engorged adult had approximately the same composition patterns as the other adults (Fig. S3). One nymph appears distinct from the rest, in terms of relative abundances; however, we cannot discern if this observation is because of the engorgement level or the variability seen in nymphs. While it is possible that amount of blood meal could affect the microbiome, future studies will need to quantify the significance and direction of such effects.

Because of the nature of tick (blood-feeding pathogens) and our sampling (ticks sent to diagnostic laboratory for pathogen detection), there is an opportunity for contamination (e.g., microbes from human skin). To address the possibility of contamination, we checked relevant literature for and verified that the major genera observed (see Fig. 5) are ecologically plausible: *Stenotrophomonas, Staphylococcus, Pseudomonas, Lysinibacillus, Lactobacillus, Klebsiella, Acinetobacter, Brevibacillus, Borrelia, and Rickettsia* have been found in either *I. scapularis* or in similar *Ixodes spp* (*Benson et al., 2004*; *Carpi et al., 2011*; *Narasimhan et al., 2014*; *Van Treuren et al., 2015*; *Abraham et al., 2017*; *Kwan et al., 2017*; *Khalaf, Mohammed & Karim, 2018*; *Kmet' & Čaplová, 2020*). We therefore believe that the major taxonomic signals derive from the tick microbiome and not from human skin contamination.

It is important to note that the ticks were crushed without surface sterilization before total DNA extraction and this is a byproduct of our collaboration as it is not necessary to surface sterilize for pathogen detection. We were not largely concerned about this for three reasons: (1) the major genera observed were all ecologically plausible, all being reported in at least one peer-reviewed study. (2) Numerous studies use ethanol washes to surface sterilize but while ethanol kills microorganisms, it does not remove the DNA, so this would minimally affect the extractions. (3) It has been suggested that surface microbes can enter the tick from the surface and colonize the gut. If bacteria from the cuticle can affect the internal microbiome, it would be important to consider the external microbiome when looking for taxa specific relationships. Therefore, destroying the DNA of these organisms using a bleach wash might hide some unknown relationships between the host and the

microbiome as well as taxa specific interactions (*Ross et al., 2018*; *Binetruy et al., 2019*). Thus, the results of this study are likely due to signals from the tick microbiome and not from noise introduced from surface contamination from human skin. We are however aware of the inflation of diversity from low biomass samples and we tried to compare the different input DNA and saw limited differences between input DNA concentrations. We also removed the low abundant ASVs and still found significant differences in alpha diversity between nymphs and adults; however, the low biomass of some of our nymphs must be considered. Our data show that we can get ecologically relevant data pertaining to tick microbiomes by collaborating with diagnostic laboratories. Moving forward it would be best to modify the protocol, including surface sterilization step and include qPCR for determining absolute abundance of the taxa present to further quantify low biomass samples.

## CONCLUSIONS

Ticks are a prominent vector for the transmission of human pathogens; however, how these pathogens are integrated as a part of the microbiome is poorly understood. We sought to determine if the presence of the TBPs were associated with differences in the microbiome. There was no significant difference in *I. scapularis* microbiome based on the presence of *B. burgdorferi*. There were also no significant differences in microbiome composition or diversity in samples with *B. miyamotoi, A. phagocytophilum, B. microti*. The number of TBPs has limited correlation to the overall diversity of the microbiome. Life stage is the most important factor associated with microbiome composition and diversity, results that are similar to work done on another pathogen vector, *I. pacificus*. Future studies should work with larger sample sizes using wild tick samples as well as investigate the functional relationship between TBPs and the tick microbiome; metagenomic and metatranscriptomic methods should be employed. That will elucidate the functional relationships that may or may not be changed depending on how many and which TBPs are present.

## ACKNOWLEDGEMENTS

The authors would like to thank Maureen Sim, for tick identification and Heather Haycock and Erica Jingozian for conducting the lab work for the Diagnostic lab, CVMDL, Pathobiology and Veterinary Science, University of Connecticut. We would also like to thank Kirsten Grond, Elizabeth Herder, Ryan Duggan, and Brent Basso for comments on the manuscript.

### Funding

This work was supported by the University of Connecticut through startup funds of Sarah Hird. The funders had no role in study design, data collection and analysis, decision to publish, or preparation of the manuscript.

## Grant Disclosures

The following grant information was disclosed by the authors:
University of Connecticut through startup funds of Sarah Hird.

## Competing Interests

The authors declare there are no competing interests.

## Author Contributions

- Joshua C. Gil analyzed the data, prepared figures and/or tables, authored or reviewed drafts of the paper, and approved the final draft.
- Zeinab H. Helal performed the experiments, authored or reviewed drafts of the paper, and approved the final draft.
- Guillermo Risatti performed the experiments, prepared figures and/or tables, and approved the final draft.
- Sarah M. Hird conceived and designed the experiments, authored or reviewed drafts of the paper, and approved the final draft.

## Data Availability

 Data are available at NCBI Short Read Archive (SRA): PRJNA666434.

## Supplemental Information

Supplemental information for this article can be found online at http://dx.doi.org/10.7717/peerj.10424#supplemental-information.

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
