# Peer review of "Ixodes scapularis microbiome correlates with life stage, not the presence of human pathogens, in ticks submitted for diagnostic testing"

_PeerJ, doi:10.7717/peerj.10424_

## Round 0.1 · original submission · Major Revisions

Both referees agreed that this is an interesting manuscript. However, they also had several concerns, in particular concerning the risk of environmental contamination of the samples. Referee 2 also suggested that there might be a bias in the comparison of the microbiome of nymphs and adult ticks, because of the difference in bacterial biomass. The referee made a series of valuable suggestions as to correct for such potential bias.

Reviewer 1 ·

Basic reporting

The authors present an interesting study on the structure of microbial communities in the black legged tick Ixodes scapularis, a species of major medical interest in North America. While the diversity of bacteria endosymbionts and tick-borne pathogens (TBP) have been extensively studied in this tick species, the authors produced novel results using field collected specimens. They notably concluded that life stages, but not TBP presence/absence, is the strongest driver of microbiome composition. I particularly appreciated the use of statistics – an approach too rarely used in this kind of metabarcoding study. The text is globally well written, and the rationale is clear. However, I have some concerns, detailed below.

Experimental design

- Materials and Methods: What about the negative controls? Controls are needed at each step of molecular protocols (extraction control, PCR control, kit control, etc). They are pivotal in metabarcoding studies because they allow the identification of bacterial contaminants that are widely present in the lab. Without these controls, it remains extremely difficult to identify contaminant bacteria, and thus to remove them from dataset before further analyses.
- By the way, rarefaction curves comparing the number of 16S reads and the number of detected bacterial taxa are not presented. Did they showed plateaued asymptotes for each library? If not, it will indicate that sequencing depth was not enough to detect all bacterial taxa present. So, it is important to show that all DNA samples have produced good quality of data.
- Materials and Methods: Some important details on tick specimens must be indicated there. The sex of adult and feeding status should be indicated in this section - not only at the end of the discussion (l.372). Most importantly, there is no detail on origins of samples: Were they from the same location? Were they collected during the same season? Over several years? Since spatiotemporal features covariate with structure of microbial communities in ticks, these details are of importance. To resume: Are these samples homogeneous from their origins? If not, it introduced some noise that deserve discussion.

Validity of the findings

- As indicated in the discussion (l.372, and not in the Materials and Methods…), specimens are engorged. So, it means some detected TBP in tick specimens may be not ‘true’ members of tick microbiota but rather transitional passengers (only present in the blood meal and unable to further infect tick tissues). Alternatively, these TBP could have been recently acquired with this last blood meal, and have simply not enough time to interact with microbes present in the gut of ticks. This may explain why there is no apparent interaction between TBP and structure of microbial communities.
- Engorged ticks may contain a lot of blood. Since the specimens used in this study were removed from humans, I expect that they were more or less engorged: substantial variation can exist between specimens. Such interindividual heterogeneity should be discussed because there are two consequences of this. First, blood is a powerful inhibitor of PCR and microbes are typically more difficult to amplify using fully engorged specimens. In this context, less 16S bacterial reads could be obtained from fully engorged specimens, introducing a bias when compared with less engorged specimens. Second, large amount of blood will massively dilute resident microbes and potentially introduce more TBP if patient was infected. Again, it may introduce another bias when compared with less engorged specimens. These variations could be included in analyses.
- Line 404-410: the authors indicate that their ticks are potentially contaminated by bacteria from human skin because they have not cleaned the cuticles before extraction. They however stated that it is not important since internal environment (gut and ovaries) must harbor far more microbes than external surfaces (cuticles). Unfortunately, this is not entirely true. Most studies aiming to characterize tick microbiome wash specimens with ethanol, buffer or bleach. A recent analysis in another American tick species showed that cuticles consistently harbor a greatest diversity of microbes than internal organs, and that these microbes are a substantial source of contamination in metabarcoding studies (Binetruy et al 2020, Parasites & Vectors). So, the authors should softened their conclusion on this aspect.
Other comments:
- Line 50: ‘obtained’ or ‘produced’ rather than ‘collected’.
- Lines 107-107: This is entirely true for hard ticks (and thus including I. scapularis) but not for soft ticks that have several nymphal stages (and thus take more blood meals during their development).

Additional comments

This work is worthy of interest. However, they are some important concerns that must be considered before acceptance.

Reviewer 2 ·

Basic reporting

Figure legends should be added. Numerous relevant references are missing.

Experimental design

More methods needed for tick collection and processing.

Validity of the findings

Results should be placed into context of well-documented 16S low-biomass sequencing artifacts. Statistical tests, sample sizes, sample details, etc should be included in figure legends.

Additional comments

The current manuscript by Gil et al examines microbial diversity associated with wild-caught Ixodes scapularis ticks, using up-to-date 16S rRNA amplicon sequencing methods and careful analysis. The authors break down their ticks into groups based on detection of tick-borne pathogens by qPCR, and find that pathogen infection is not associated with an altered microbiome. In contrast, they find that life stage is associated with differences in alpha and beta diversity. The paper is well written, and the figures are well put together and clear. Overall, while there have been a number of tick microbiome papers in recent years, the question of differences in the microbiome as it relates to specific pathogen infection is unresolved and quite interesting.

Major comments:
- More detail is required on methods regarding handling/storage of ticks prior to DNA extraction. Were the ticks alive or dead, and if dead for how long and stored under what conditions? These points could potentially influence the viability and abundance of tick-associated bacteria. Were ticks washed or externally sterilized prior to DNA extraction? This has commonly been performed in other papers to remove external contaminants. If washing was not performed, this needs to be acknowledged and the interpretation of detected microbes being internal needs to be revised.

- I am concerned about the finding of higher diversity in nymphs. Nymphs are much smaller than adult ticks, and therefore the bacterial biomass associated with nymphs is correspondingly less than that of adults. There is a well-documented effect of biomass/input DNA on diversity metrics in the broader microbiome literature (see de Goffau et al 2018 Nature Microbiology; Erb-Downward et al 2020 mBio; Salter et al 2014 BMC Biology; and other relevant papers cited by these works. Also see Hammer et al 2019 FEMS Microbiology Letters for a discussion of best practices for microbiome analysis of low-biomass arthropod samples.). I am therefore concerned that the major finding of this paper (that diversity decreases with life stage) is an artifact of differences in input bacterial biomass between lifestages. Proposed methods for assessing/dealing with low biomass may not be feasible with the small nymphs used examined in this study (e.g. visual detection of bacteria via fixation, FISH, and imaging). The authors could use qPCR and known standard to roughly quantify bacterial copy number in nymphs (see Fig 1 Hammer et al PNAS 2017 for an example of this, also Couper 2019). Minimally, I would like to see the authors acknowledge and discuss this well-documented biomass-deep sequencing phenomenon in the Discussion.

- The authors should provide detailed figure legends for each figure that explain (among other things) the sample sizes, statistical tests performed, etc. E.g. for Figure 1, are the data combined nymphs and adults?

- Check spelling throughout, e.g.:
o Table S4, “Analasma”
o Figure 2, “diveristy”

- The authors should cite a number of significant and relevant published references which are missing from this manuscript and are important for placing this work within the current state of knowledge:

o Landesman et al 2019 FEMS Microbiology and Ecology; Landesman et al 2019 Ticks and Tick Borne Diseases. Both of these papers examine Eastern US-collected I. scapularis nymphal ticks, with the FEMS paper looking at over 400 samples.
o Couper et al 2019 Ecology and Evolution. Found low diversity in Ixodes microbiomes (7 OTUs). Methods for reducing false signal from contamination.
o Ross et al 2017 ISME found that alpha diversity differences between adult ticks (in particular between female and male ticks) were correlated with total bacterial 16S abundance by qPCR. This finding is in line with many studies demonstrating that low input biomass leads to artificially high alpha diversity driven by contaminants.

---

## Round 0.2 · accepted · Accept

The referees were happy with the revised version of the manuscript. Please check the last pending points (typos and figures).

Reviewer 1 ·

Basic reporting

The authors responded to all reviewers' comments and modified the manuscript accordingly. Thank you for that. There are several typos to correct, eg at lines 81 (‘Babesia’), 113 (dot and space). Please also make clear that the microorganisms B. microti and B. miyamotoi do not belong to the same lineage, eg using term such as Ba. microti and Bo. miyamotoi.

Experimental design

no comment

Validity of the findings

no comment

Additional comments

no comment

Reviewer 2 ·

Basic reporting

The authors have expanded their citation of the literature in this resubmission, as recommended in the first review.

Experimental design

The authors have nicely revised their original manuscript with attention to reviewer concerns pertaining to low biomass samples.

Validity of the findings

No comment

Additional comments

The authors have done a nice job revising their manuscript in response to reviews. I think it is very much improved and should be accepted. I have a minor comment. Supplemental figure 3 (which is nice) has overlapping colors for some of the taxa (Klebsiella-Rickettsia, Family_comamonadaceae-Sphingomonas-Stenotrophomonas, etc). If possible I recommend expanding the color palette to allow for more differentiation, perhaps by using less "hot" colors. Finally, I was unable to see find the supplemental figure legends.